# Neurogenic vs. Myogenic Origin of Acquired Muscle Paralysis in Intensive Care Unit (ICU) Patients: Evaluation of Different Diagnostic Methods

**DOI:** 10.3390/diagnostics10110966

**Published:** 2020-11-18

**Authors:** Humberto D.J. Gonzalez Marrero, Erik V. Stålberg, Gerald Cooray, Rebeca Corpeno Kalamgi, Yvette Hedström, Bo-Michael Bellander, Inger Nennesmo, Lars Larsson

**Affiliations:** 1Section of Clinical Neurophysiology, Department of Clinical Neuroscience, Karolinska Institutet, 171 77 Stockholm, Sweden; humberto.skott@sll.se (H.D.J.G.M.); gerald.cooray@ki.se (G.C.); 2Department of Clinical Neurophysiology, Section of Neuroscience, Uppsala University, 751 85 Uppsala, Sweden; stalberg.erik@gmail.com; 3Department of Physiology and Pharmacology, Karolinska Institutet, 171 77 Stockholm, Sweden; rebeca.corpeno.kalamgi@ki.se (R.C.K.); yvette.hedstrom@ki.se (Y.H.); 4Section of Neurosuregery, Department of Clinical Neuroscience, Karolinska Institutet, 171 77 Stockholm, Sweden; bo-michael.bellander@sll.se; 5Department of Pathology, Karolinska University Hospital, 171 64 Stockholm, Sweden; inger.nennesmo@sll.se; 6Department of Biobehavioral Health, the Pennsylvania State University, University Park, PA 168 02, USA

**Keywords:** critical care, myosin, myopathy, ENeG, EMG, CMAP, muscle biopsy

## Abstract

**Introduction**. The acquired muscle paralysis associated with modern critical care can be of neurogenic or myogenic origin, yet the distinction between these origins is hampered by the precision of current diagnostic methods. This has resulted in the pooling of all acquired muscle paralyses, independent of their origin, into the term Intensive Care Unit Acquired Muscle Weakness (ICUAW). This is unfortunate since the acquired neuropathy (critical illness polyneuropathy, CIP) has a slower recovery than the myopathy (critical illness myopathy, CIM); therapies need to target underlying mechanisms and every patient deserves as accurate a diagnosis as possible. This study aims at evaluating different diagnostic methods in the diagnosis of CIP and CIM in critically ill, immobilized and mechanically ventilated intensive care unit (ICU) patients. **Methods**. ICU patients with acquired quadriplegia in response to critical care were included in the study. A total of 142 patients were examined with routine electrophysiological methods, together with biochemical analyses of myosin:actin (M:A) ratios of muscle biopsies. In addition, comparisons of evoked electromyographic (EMG) responses in direct vs. indirect muscle stimulation and histopathological analyses of muscle biopsies were performed in a subset of the patients. **Results**. ICU patients with quadriplegia were stratified into five groups based on the hallmark of CIM, i.e., preferential myosin loss (myosin:actin ratio, M:A) and classified as severe (M:A < 0.5; *n* = 12), moderate (0.5 ≤ M:A < 1; *n* = 40), mildly moderate (1 ≤ M:A < 1.5; *n* = 49), mild (1.5 ≤ M:A < 1.7; *n* = 24) and normal (1.7 ≤ M:A; *n* = 19). Identical M:A ratios were obtained in the small (4–15 mg) muscle samples, using a disposable semiautomatic microbiopsy needle instrument, and the larger (>80 mg) samples, obtained with a conchotome instrument. Compound muscle action potential (CMAP) duration was increased and amplitude decreased in patients with preferential myosin loss, but deviations from this relationship were observed in numerous patients, resulting in only weak correlations between CMAP properties and M:A. Advanced electrophysiological methods measuring refractoriness and comparing CMAP amplitude after indirect nerve vs. direct muscle stimulation are time consuming and did not increase precision compared with conventional electrophysiological measurements in the diagnosis of CIM. Low CMAP amplitude upon indirect vs. direct stimulation strongly suggest a neurogenic lesion, i.e., CIP, but this was rarely observed among the patients in this study. Histopathological diagnosis of CIM/CIP based on enzyme histochemical mATPase stainings were hampered by poor quantitative precision of myosin loss and the impact of pathological findings unrelated to acute quadriplegia. **Conclusion**. Conventional electrophysiological methods are valuable in identifying the peripheral origin of quadriplegia in ICU patients, but do not reliably separate between neurogenic vs. myogenic origins of paralysis. The hallmark of CIM, preferential myosin loss, can be reliably evaluated in the small samples obtained with the microbiopsy instrument. The major advantage of this method is that it is less invasive than conventional muscle biopsies, reducing the risk of bleeding in ICU patients, who are frequently receiving anticoagulant treatment, and it can be repeated multiple times during follow up for monitoring purposes.

## 1. Introduction

Severe muscle wasting and impaired muscle function accompany critical illness in intensive care unit (ICU) patients with negative consequences for recovery from primary disease and weaning from the respirator. Several studies show unambiguously that neuromuscular dysfunction, resulting in muscle wasting and weakness, is the most persistent and debilitating problem for survivors from the ICU for several years after hospital discharge [1,2]. There is accordingly a significant need for more research focused on the mechanisms underlying the muscle wasting and weakness in ICU patients [3]. Primary disease, sepsis and multi-organ failure undoubtedly contribute to the impaired muscle function, but there is heterogeneity of underlying disease and pharmacological treatment among patients with similar outcomes. Thus, it is highly likely that the common components of ICU treatment per se, such as muscle unloading, mechanical ventilation, neuromuscular blockade, sepsis, and/or corticosteroids are directly involved in the progressive impairment of muscle function during long-term ICU treatment.

There are different etiologies underlying acquired muscle wasting and impaired muscle function in ICU patients such as acquired myopathy, initially named acute quadriplegic myopathy before critical illness myopathy (CIM) became the most accepted term. The pathophysiological processes causing acquired quadriplegia in critically ill ICU patients has been shown to involve myopathic and/or neuropathic processes, resulting in a spectrum of physiological dysfunctions. Moreover, the natural cause of critical illness is highly dependent on the underlying cause. There are also several specific forms such as Guillain–Barré syndrome and sepsis-induced myopathy [4]. Myogenic dysfunction, as a cause of critical illness, was—for many years—not identified and erroneously assumed to be caused by inflammatory (e.g., Guillain–Barré syndrome) or non-inflammatory neurogenic dysfunction (i.e., critical illness polyneuropathy, CIP) [5]. The inclusion of CIM among neuropathies is, in part, due to misinterpretations of electroneurographic (ENeG) and electromyographic (EMG) recordings that may mimic neurogenic lesions due to altered muscle membrane excitability in CIM [5,6,7]. CIM was initially regarded as a rare condition of limited clinical significance, but we now know that neuromuscular dysfunction is found in up to 30% of the general ICU population and in 70–100% of certain sub-groups [8,9]. This potentially lethal condition prolongs the recovery of critical care patients, thereby increasing the median ICU treatment costs by three times [10,11]. Additional substantial costs are associated with the subsequent extended rehabilitation requirements and drastically impaired quality of life. The current respiratory disease caused by the novel coronavirus SARS-CoV-2, also known as Corona Virus Disease 2019 (COVID-19), has put significant strains on intensive critical care and the need for assisted ventilation [12,13]. In a previous study on neuro-ICU patients who were mechanically ventilated for an average of nine days with controlled mechanical ventilation, due to insufficient central ventilatory drive, all patients showed preferential myosin loss, i.e., the hallmark of CIM [14,15,16]. A similar preferential myosin loss was reported in experimental studies where rats were exposed to long-term controlled mechanical ventilation [14,15,16]. Assisted mechanical ventilation is preferred in the ICU setting to reduce ventilator-induced lung injury (VILI). The proportion of patients developing CIM is, in our experience, lower in patients exposed to assisted rather than controlled mechanical ventilation. However, a large proportion of COVID-19 ICU patients are exposed to controlled mechanical ventilation, especially if ventilated in the prone position. Accordingly, there is reason to believe that VILI and the systemic release of factors from lungs have a negative effect on peripheral organs, including muscle, suggesting that VILI may play an important role in CIM pathophysiology. This is consistent with the dramatic increase in ICU patients with CIM during the current COVID-19 pandemic.

Today, ICU patients often fail to receive a correct diagnosis and ICU patients with acquired muscle paralysis/weakness are frequently conjoined into the Intensive Care Unit Acquired Muscle Weakness (ICUAW) entity, sometimes justified by the assumption that underlying etiological or pathophysiological processes are of limited clinical significance. However, this is in sharp contrast to the robust efforts being made in order to understand the pathophysiological processes causing other neuromuscular disorders, as this is imperative for the understanding of underlying mechanisms and the development of efficient intervention strategies. The distinction between CIM and CIP is important for prognosis and recovery rate, especially after ICU discharge. Furthermore, early physical therapy is of significant importance for the recovery from CIM since complete mechanical silencing (a lack of external strain caused by weight bearing and internal strain related to the activation of contractile proteins), unique to deeply sedated or pharmacologically paralyzed mechanically ventilated ICU patients, is an important factor triggering CIM [14,17,18,19], but is probably of less importance for CIP. In addition, novel pharmacological intervention strategies have recently been evaluated in experimental ICU models and are currently being translated into clinical investigations [20,21,22,23], i.e., interventions targeting underlying mechanisms and thus emphasizing the need for accurate diagnosis. Accordingly, there is a compelling need for improved diagnosis and monitoring of CIM and CIP in the ICU setting and the preferential loss of the molecular motor protein myosin and myosin-associated proteins represent a hallmark of CIM. The aim of this study is to compare commonly used histopathological analyses of muscle tissue, together with standard and advanced electrophysiological methods used in the diagnosis of CIM/CIP, and how they are related to the hallmark of CIM, i.e., preferential myosin loss, to improve the differential diagnosis between CIM and CIP.

## 2. Materials and Methods

### 2.1. Patients

In this retrospective study, patients on mechanical ventilation in the ICU were screened for preliminary inclusion, i.e., that they had an acquired general muscle paralysis of upper and lower limb muscles [24]. For point of inclusion, they should also have received a minimum of five days of mechanical ventilation and undergone neurophysiological tests, one of which must have been during the time when they required mechanical ventilation, and must have had a muscle biopsy taken at the same time for the measurement of the myosin:actin (M:A) ratio. Participants’ age, gender, days since onset of symptoms, total time on respirator and underlying diseases are summarized in Table 1. This study was not designed to investigate the association between underlying disorders and acquired muscle paralysis, but we could not detect a significant relation between preferential myosin loss and underlying disorders in the 144 ICU patients included in this study (infection, multiorgan dysfunction, cerebral insult, malignancy, neurological disorders, trauma and transplantation). The patients were recruited from Uppsala University Hospital, Karolinska University Hospital (Solna and Huddinge) and Södersjukhuset in Stockholm. The study was approved by the local ethics board at Karolinska Hospital (Dnr 96-341, Dnr 2016/242-31/2) and Uppsala University Hospital (Dnr2008/174, Dnr 2009/124).

### 2.2. Biopsy Techniques

Small muscle samples (>80 mg) are typically obtained with the percutaneous conchotome biopsy instrument. Although the percutaneous muscle biopsy technique is a minor intervention useful for diagnostic purposes of neuromuscular disease, it is not used in daily monitoring. An alternative small 14-gauge 11 cm long soft-tissue semi-automated biopsy disposable needle instrument (Temno Evolution^®^), originally designed for kidney, liver, lung, thyroid or breast biopsies, was therefore evaluated for diagnostic and monitoring purposes. In order to evaluate the reliability and precision of this method, M:A ratios have been compared between duplicate biopsies and with results from open muscle biopsies in a porcine experimental ICU model [25,26]. In addition, M:A ratios have been compared between the small biopsies and those obtained with the percutaneous muscle biopsy method in ICU patients. An initial attempt was also made to use an aspiration biopsy technique due to promising results under experimental condition. However, this method proved less valuable in the clinical situation (see Results).

### 2.3. Biochemical Analyses of Myosin Expression

Actin and myosin quantification was determined by 12% sodium dodecyl sulphate–polyacrylamide gel electrophoresis (SDS–PAGE) on two 10 µm cryo-cross-sections from percutaneous muscle biopsies or from approximately half of the fine needle biopsy. The acrylamide concentration was 4% (w/v) in the stacking gel and 12% in the running gel, and the gel matrix included 10% glycerol. Samples of 5 μl were loaded together with 5 μL of standard dilutions. Electrophoresis was performed at 32.0mA for 5 h with a Tris–glycine electrode buffer (pH 8.3) at 15 °C (SE 600 vertical slab gel unit; Hoefer Scientific Instruments). The gels were stained using SimplyBlue SafeStain (Invitrogen) and subsequently scanned in a soft laser densitometer (Molecular Dynamics, Sunnyvale, CA, USA) with a high spatial resolution (50 μm pixel spacing) and 4096 optical density levels. The volume integration function was used to quantify the amount of protein on 12% and 6% gels (ImageQuant TL Software v. version 3.3, Amersham Biosciences, Uppsala, Sweden).

The subcellular fractionation of cytosolic vs. myofibrillar myosin was evaluated using the Cell Compartment Kit for subcellular fractionation of intact cells and tissues (Qiagen, Sollentuna, Sweden).

Histopathological evaluation of CIM was based on enzyme histochemical myosin ATPase or immunocytochemical stainings using myosin heavy chain antibodies according standard procedures.

### 2.4. Routine Neurophysiology

#### 2.4.1. Neurography

Motor and sensory neurography was performed in all patients. Amplifier filters were set to 20 Hz and 10 KHz for all neurography studies. The temperature was kept above 29 °C on the dorsum of the hand and above 27 °C on the dorsum of the foot. Motor conduction velocity (MCV) was calculated using distal and proximal stimulation with the measurement of compound muscle action potential (CMAP) in the median, tibial and fibular nerve. Distal stimulation was performed 80 mm from the target muscle. The amplitude of CMAP was measured between the baseline and the negative peak. Duration was measured from the onset latency of the CMAP to the first negative–positive baseline crossing. The drop in amplitude and dispersion of the CMAP when comparing responses obtained after proximal and distal stimulation was assessed. F-responses (latency and persistence) were not analyzed as the values were too uncertain in this patient material, where low CMAP amplitudes resulted in low F-response persistence. Furthermore, since persistence also depends on limb inactivity [27], the interpretation of persistence as a measure of conduction block would be inadequate for the present study. Sensory neurography was performed with surface electrodes both for stimulation and recording. Median, ulnar, radial and sural nerves were studied, usually on one side, but sometimes bilaterally. The tests were performed with 14 cm distance between stimulating and recording electrodes. Median and ulnar nerves were studied ortodromically, while sural and radial nerves were studied antidromically. Sensory conduction velocity (SCV) was calculated using the onset latency of the averaged response.

#### 2.4.2. Neurography Reference Values and Indices

Neurography values were referenced to healthy reference values previously measured at our laboratories [28,29,30]. These values have been regressed on age and height using linear regression analysis with the above-mentioned factors as covariates. Deviations from reference values were described in number of standard deviations (Z-score). The mean of the normal reference values were set to 0, while the 99.9% confidence limits were approximated to lie within ±3 Z-scores. We decided to use a wide variation in normality due to the hostile environment in the ICU. Sensory amplitudes were logarithmically transformed before regression analysis with age and height. The lower range for sensory nerve action potentials was defined at 0,2 uV to ensure the definition of the logarithm of the data. The lower range for CMAP data was 0, as these values were not logarithmically transformed.

The degrees of abnormality of neurography parameters were estimated using indices for CMAP, sensory nerve action potential (SNAP), motor nerve conduction velocity (MCV) and sensory nerve conduction velocity (SCV). The indices were calculated using a modified algorithm of a previously published method [2]. They were estimated as follows:CMAP index: ((Z-score CMAP median amplitude (left) + Z-score CMAP median amplitude (right) + Z-score CMAP ulnar amplitude (left) + Z-score CMAP ulnar amplitude (right) + Z-score CMAP fibularis amplitude (left) + Z-score CMAP fibular amplitude (right)+ Z-score CMAP tibial amplitude (left) + Z-score CMAP tibial amplitude (right))/√*n*. (*n* = number of values per patient)
Sensory nerve action potential (SNAP) index: (z-score median amplitude digigitorum III (dig. III or long finger, left) + z-score median amplitude dig III (right) + ulnar dig V amplitude (left) + ulnar dig. V (little finger) amplitude (right) + radial amplitude (left) + radial amplitude (right)+ sural amplitude (left) + sural amplitude (right) )/√*n* (*n* = number of values per patient)

Motor conduction velocity (MCV) index and sensory conduction velocity (SCV) index were estimated in a similar way.

Healthy subjects were used to get normative values for the indices and only 5% of the subjects had index values below −3. The stratification of abnormality at −3 resulted in a *p*-value of 0.05.

#### 2.4.3. Intramuscular Stimulation and Recording

Intramuscular stimulation and recording were performed in some of the patients. Details of the methods have been described previously [21]. The Tibialis anterior muscle (TA) was used for recording. The skin temperature over the TA muscle was kept at >32 °C. The needle electrodes were Teflon-coated monopolar needle electrodes (28G, 30 mm (cathode) and 15 mm (anode)), each with a surface area of 0.35 mm^2^ (Alpine BioMed Asp Instrument, Skovlunde, Denmark). The cathode was inserted to a depth of about 20 mm perpendicular to the skin surface, and the anode less deep. The surface electrode, used as an anode, was a non-polarizable gel surface electrode (Blue Sensor Tab; Ambu A/S, Ballerup, Denmark). The effect of stimulation position was also tested. We kept the cathode in the distal third of the TA muscle. The cathode was held in a constant position that was optimized to give the highest CMAP, whereas the anode was moved to different test positions. Stimulation strength was increased slowly until supramaximal stimulation was achieved (from 1 to 100 mA, duration 0.1 ms). Nicolet VikingSelect software, version 11.1 (CareFusion Middleton, Wisconsin) was used. The amplifier bandwith was 2 Hz to 5 kHz for maximal nerve stimulation compound muscle action potential (neCMAP), whereas a bandwidth of 20 Hz to 5 kHz was used for refractory measurements. The other test settings were as follows: stimulus duration—0.1 ms; sweep speed—2–5 ms/division; amplification—between 100 μV and 10 mV/division; and stimulation rate—1 Hz [21].

#### 2.4.4. Ratio of Muscle Response after Muscle and Nerve Stimulation, Respectively

First, both intramuscular recording and stimulating MNEs were positioned until a maximal response was obtained (Figure 1). Keeping this recording position, the fibular nerve was then stimulated supramaximally at the level of the fibular head. Both direct muscle stimulation and nerve stimulation tests were performed between 3 and 5 times in each subject and the mean value was calculated and reported. The stimulus duration was 0.1 ms, and stimulation rate was 1 Hz [31]. A ratio of peak-to-peak amplitudes obtained with nerve and muscle stimulation, respectively, was calculated—the ne/dm index [8,31].

#### 2.4.5. Refractory Period

The refractory period was estimated in a number of patients. To estimate the refractory period of the muscle fibers, the stimulus strength was set to give a submaximal response for intramuscular stimulation [31]. Double stimulation with varying interstimulus intervals (0–20 ms in 1 ms steps) was used. We defined the refractory period to be the interstimulus interval that gave a response amplitude which was half that of the single stimulus response. Only simple responses were accepted. Amplitude measurements were usually performed between baseline and negative peaks. In cases with a superimposition of the test signal with earlier components from the first stimulation, the negative to positive peak was used. In the case of complex signals, amplitude measurements are not meaningful. In these signals, it was usually possible to measure the refractory period for individual components, but this does not correspond to the criterion of 50% amplitude loss. The refractory period values from these signals were not included in the material (Figure 2).

#### 2.4.6. Electromyography

Conventional concentric needle electrodes were used. Most of the studies were performed with Nicolet VikingSelect equipment (CareFusion Middleton WI, USA). Some of the recordings were made with KP-classic (Natus Medical Incorporated, San Carlos, CA, USA). Proximal and distal muscles were examined in arms and legs. Spontaneous activity was assessed in at least ten positions in each muscle. The spontaneous activity was scored as follows; 0 = no spontaneous activity in any of the studied muscled, 1 = spontaneous activity in less than 5 out of 10 sites in at least one muscle, 2 = spontaneous activity in 5–10 positions in one muscle, 3 = spontaneous activity in 5–10 positions in 2 muscles, 4 = spontaneous activity in 5–10 positions in more than 2 muscles. If the patient could activate some muscles, the analysis focused on the recruitment pattern (early-late), degree of polyphasicity, amplitudes and fullness if good effort was obtained.

### 2.5. Statistics

Statistical calculations were performed using MATLAB (Nantick, MA, USA). Mean, standard deviation, correlation coefficients (Pearson correlation) and ANOVA F-test estimates were calculated using a standard formula. Tukey’s range test was used to identify individual differences between the groups.

## 3. Results

### 3.1. Patients

A total of 144 ICU patients who developed general weakness/paralysis of the limbs and trunk muscles during critical care were included in the study. All ICU patients had been mechanically ventilated for five days or longer at the time of percutaneous muscle biopsy and electrophysiological examination (range 5–165 days, Table 1).

### 3.2. Electrophoretic Separations of Myofibrillar Proteins

The preferential loss of myosin and myosin-associated proteins represents the hallmark of CIM and separates it from CIP and other types of acquired muscle paralyses in the ICU [5,32]. Myosin content was normalized to actin content on Coomasie-stained 12% SDS–PAGE and compared with one to two controls run on the same gel at five different protein concentrations.

The myosin:actin (M:A) ratio varied between 0 and 2.2 in the ICU patients and between 1.9 and 2.4 in controls. The M:A ratio was not significantly affected by the duration the patient had been mechanically ventilated and immobilized for. In accordance with experimental studies [33], we have previously confirmed a slight preferential myosin loss in old age, i.e., in 55 healthy men and women ranging in age between 24 and 94 years [34,35]. However, this decline in the M:A ratio was primarily observed in very old individuals. In the present clinical material, the two oldest patients (87 and 94 years) were not included, i.e., 142 patients were included in the analyses. All ICU patients younger than 85 years were included, since there was no statistically significant age-related decline in the control group in the 24–85 age range (Appendix A). Based on these data, an M:A ratio of 1.7 was chosen as the lower reference value (10–90% confidence limit) and patients were divided into five groups based on this ratio: severe (M:A < 0.5; *n* = 12), moderate (0.5 ≤ M:A <1; *n* = 40), mildly moderate (1 ≤ M:A <1.5; *n* = 49), mild (1.5 ≤ M:A < 1.7; *n* = 24) and normal (1.7 M:A; *n* = 19).

In a previous study, the M:A ratio was reported to be affected by the total amount of protein loaded onto the gel [36]. In the current study, M:A ratios were therefore compared at five different protein concentrations on Coomassie-stained 12% SDS–PAGE in all patients (*n* = 142) showing different degrees of preferential myosin loss and in healthy controls (*n* = 47). The M:A ratios did not differ between the five different protein concentrations, varying from 20 to 100% of the original muscle biopsy protein concentration, using the current protocol for separation and quantification of myofibrillar proteins (Appendix A).

In experimental studies, two aspiration muscle biopsies and one open muscle biopsy were obtained from 16 pigs to evaluate the reliability and precision of the M:A ratios determined from Coomassie-stained 12% SDS–PAGE. Prior to muscle biopsy, all pigs had been mechanically ventilated for five days [26,37,38]. The M:A ratios did not differ between either the open biopsy (2.05 ± 0.03), the first (2.08 ± 0.06) or the second (2.05 ± 0.07) aspiration muscle biopsies, indicating high reliability and precision. The aspiration biopsies were taken in close proximity to the region from where the open muscle biopsy was obtained, i.e., from an area where the muscle was exposed and could easily be identified without penetrating skin and subcutaneous tissue [26,37,38]. When the aspiration biopsy method was tested in parallel with percutaneous muscle biopsies for diagnostic purposes in ICU patients, more than 50% of the samples did not contain muscle tissue. The remaining aspiration biopsies contained blood, subcutaneous tissue and muscle, resulting in poor correlation between the myosin:actin ratio from percutaneous and aspiration biopsies. An alternative small soft-tissue semi-automated disposable biopsy needle instrument was therefore evaluated for diagnostic purposes. After a short (~1 mm) skin incision, a 14 Gauge 11 cm long biopsy needle (Figure 1) without coaxial introducer was introduced into the muscle and small (4–15 mg) muscle biopsies, free from subcutaneous tissue, were obtained from 12 ICU patients. M:A ratios varied between 0.9 and 2.2 and identical ratios were observed in samples obtained with the conchotome biopsy instrument (1.7 ± 0.1) and the semi-automated needle (1.7 ± 0.1) instruments. A strong linear relationship (r^2^ = 0.98; *p* < 0.001) was accordingly observed between the M:A ratios obtained with the two different biopsy instruments (Figure 3).

In a subsample (*n* = 8), the myofibrillar vs. cytoplasmic content of myosin was evaluated in controls with a normal myosin:actin ratio (2.1–2.3, *n* = 3) as well as in patients with a significant preferential myosin loss (myosin:actin ration 0.4–0.6, *n* = 5). There were no significant differences in the relative myofibrillar and cytoplasmic myosin contents between the two groups (75 ± 9% in controls and 75 ± 4% in patients with preferential myosin loss). Thus, there was no evidence of a disproportionate amount of myosin in the cytoplasmic vs. myofibrillar pool of myosin in relation to the preferential myosin loss.

### 3.3. Muscle Histopathology

Morphological and histopathological examinations have frequently been used in the diagnosis of CIM and based on loss or blurring of myosin content based on enzyme or immunohistochemical stainings. We therefore compared histopathological examinations of muscle biopsies with the M:A ratios in 51 ICU patients in the diagnosis of CIM. None of the patients with normal M:A ratios were diagnosed with CIM. The number of patients diagnosed with CIM based on histopathology increased with decreasing M:A ratios. However, many patients were not diagnosed with CIM in spite of a pathological preferential myosin loss, even when myosin loss was moderate to severe (Figure 4). Patients were diagnosed with CIP based on the presence of a fiber type grouping and grouped atrophy in the muscle sections. However, both fiber type grouping and grouped atrophy reflect a chronic denervation**–**reinnervation process by far preceding the acute generalized muscle weakness, confirming that many ICU patients have a long clinical history, including neuropathies, prior to ICU admission.

### 3.4. Electrophysiology

The general standard electrophysiology (EDX) profile is summarized in Table 2. The obtained data are compared with values from healthy age-matched subjects [32] for all parameters except for CMAP and SNAP indices, which were estimated using the z-scored values of the motor and sensory nerve conduction parameters (see Materials and Methods).

Sensory and motor neurographies were performed in all patients according to the standard protocols of the laboratories. There were significant correlations between CMAP index and SNAP index (correlation coefficient of 0.53; *p* < 0.001) and between SNAP index and SCV (correlation coefficient of 0.46; *p* < 0.001).

CMAP durations were obtained from the abductor pollucis brevis muscle (APB), abductor digiti minimi muscle (ADM), extensor digitortum brevis muscle (EDB) and abductor hallucis muscle (AH), while recordings without distinct CMAPs were excluded (*n* = 21). Due to very low amplitudes from EDB and AH in 60 patients, the results are given for ABP and ADM. The CMAP durations were significantly (*p* < 0.001) longer in patients than in controls (controls: APB 5.03 ± 0.68, N = 765, limit 6.39; ADM 5.14 ± 0.61, N = 698, limit 6.37) for APB (z-score 1.8 ± 2.8; 6.25 ± 1.9 ms) and ADM (z-score 2.3 ± 2.6; 6.5 ± 1.7 ms). Statistically significant correlations were observed between both ADM duration (r = −0.36; *p* < 0.001) and APB duration (r = −0.20; *p* < 0.05) and the M:A ratio, but correlations were weak (Appendix A).

### 3.5. Direct Muscle Stimulation in the Tibialis Anterior Muscle

Recordings were obtained from 71 patients. The direct muscle stimulation CMAPs (dmCMAPs) were polyphasic and had prolonged durations in 88% of the patients. Both dmCMAP and nerve stimulation CMAP (neCMAP) amplitudes showed a weak correlation with the M:A ratio (r = 0.3; uncorrected *p* < 0.01), A significant correlation was observed between dmCMAP and neCMAP (r = 0.7; corrected *p* < 0.001). The refractory period (five patients with accepted responses) was 6.6 ± 3.6 ms (range 5–13 ms), i.e., abnormal in all five patients. There was no correlation between refractory period and dmCMAP. In the control material, the refractory period was 2.5 ± 0.6 ms [28].

The ratio mean values of direct vs. indirect CMAP amplitude responses varied considerably from one test to another and were compared with M:A ratios in 26 patients. The ratio values indicated CIM in both patients with an M:A ratio less than 0.5, and in patients with higher M:A ratios. Surprisingly, three patients with normal M:A ratios had direct vs. indirect stimulation results indicating CIM. Furthermore, direct vs. indirect stimulation ratios indicated CIP in patients with M:A ratios that varied between 1.0 and 1.9 (Figure 5). One of them was characterized as suspected CIM (M:A ratio 1.2) and the others as not having CIM. The majority of patients with a myosin:actin ratio above 1.0 were normal according to direct vs. indirect stimulation. The ratio of muscle response amplitudes after nerve and muscle stimulation, respectively, varied considerably from one test to the next and three—or sometimes five—different tests had to be carried out for each patient. The variability of the results may also be the reason for the inconsistent results when compared with M:A ratios (Figure 6). The ratio was calculated as the mean for each patient, which varied in the entire material between 0 and 2.1. In controls, the 95% confidence limit of the nerve:muscle ratio was 0.4–2.7.

### 3.6. Electromyography

Sixty-three patients out of 142 could not activate their muscles and, in those, only degree and type of spontaneous activity could be analyzed according to the scoring given above (Table 3).

Fibrillations and positive sharp waves were recorded in 87–100% of the patients in the different myosin loss groups (Table 4). In 81 patients, sufficient voluntary activity could be recorded to allow visual analyses of single motor unit potentials. The findings were interpreted as myopathic in 40, neurogenic in 7, mixed in 6 (short duration, low or high amplitudes, polyphasic motor unit potentials (mups) with late recruitment) and normal in 17. There were more “myopathic” EMG findings in patients with the lowest (0–1.5) myosin:actin ratios (46% compared to 16% in the group with myosin:actin ratio above 1.7). In general, there were more abnormalities in leg muscles than in arm muscles, but no proximal–distal gradient was observed.

### 3.7. M:A Ratio, Electrophysiology and Duration of Mechanical Ventilation

Patients were classified into five groups depending on myosin loss (see above) and electrophysiological and clinical parameters were compared between groups using one-way ANOVA. In none of the 142 patients was an inexcitable muscle membrane recorded, but the response amplitude on direct muscle stimulation was lower and the CMAP duration longer in patients showing severe preferential myosin loss, but not consistently in the other groups (Table 4, Figure 7). We also stratified patients in each group using the EDX estimates according to the following rules: CMAP index and SNAP index less than −3, only CMAP index less than −3 and both indices within reference limits (Figure 8). The presence of a low CMAP index and normal SNAP index is seen to increase as the myosin concentration drops.

## 4. Discussion

Acquired paralysis is frequently observed in mechanically ventilated ICU patients exposed to long-term mechanical ventilation and is characterized by a general loss of upper and lower limb muscle function, and intact cognitive and sensory function, while craniofacial muscle functions are spared or less affected. The muscle paralysis and severe muscle wasting may be caused by a spectrum of different underlying processes, but it has become increasingly evident that primary myogenic dysfunction is the most common pathophysiological mechanism underlying the acquired muscle paralysis, i.e., the critical illness myopathy (CIM, [5,7]). Diagnosis of acquired muscle paralysis has traditionally been based on routine electrophysiological examinations, but conventional ENeG and EMG do not reliably separate myogenic (CIM) and neurogenic (CIP) origins. This has, for many years, resulted in an overestimation of CIP or the pooling of all ICU patients with acquired muscle paralysis into “Intensive Care Unit Acquired Weakness” (ICUAW). This is unfortunate for several reasons since CIM and CIP have different prognoses and rates of recovery. Diagnostic tests should aim at unraveling the underlying mechanisms, and interventions should target pathophysiological mechanisms.

To improve the diagnostic precision of electrophysiological methods in the differentiation between CIM and CIP, Rich and co-workers introduced an elegant method where the EMG amplitude response upon direct muscle stimulation was compared with indirect stimulation via the motor nerve [39]. CIM would then be characterized by a low-amplitude response upon both direct and indirect stimulation, while CIP would show a low nerve stimulation response and a normal muscle response. This is a very attractive and interesting approach, but is technically challenging and time-consuming [32]. Similar to conventional ENeG and EMG, there is an association between the CIM response to direct vs. indirect stimulation and preferential myosin loss, but the precision is low. Low indirect vs. direct muscle stimulation both strongly support a neurogenic lesion, but this was rarely observed among the patients included in this study. In addition, it is not known if this neurogenic response represents a pre-existing neuropathy acquired prior to ICU admission or during critical care. As expected in a general ICU population, pre-existing neurogenic lesions were observed according to histopathological analyses of muscle biopsy specimens demonstrating muscle fiber type groupings and grouped atrophy, which are signs of a chronic peripheral denervation–reinnervation process that far precedes the acute acquired muscle paralysis in ICU patients.

Pathological spontaneous EMG activity (fibrillation potentials and positive sharp waves) was observed in a large majority of the ICU patients included in this study and was unrelated to myosin loss. Fibrillation potentials and positive sharp waves are typically recorded approximately three weeks after peripheral denervation with motor axon loss, but also in response to muscle membrane damage. In ICU patients, we have frequently observed spontaneous pathological EMG activity within the first two weeks after ICU admission and mechanical ventilation, making a primary muscle membrane defect more likely than motoneuron loss during the initial days of critical care. Thus, spontaneous EMG activity may be more sensitive in the diagnosis of CIM than myosin loss or a decreased CMAP amplitude. Alternatively, the spontaneous EMG activity recorded in the ICU patients need not be primarily related to CIM, but only to common factors associated with modern critical care. Propofol is the most commonly used sedative in adult patients under critical care who are in need of mechanical ventilation. However, propofol is also known to cause low-grade myotoxicity, which can lead to propofol infusion syndrome, involving rhabdomyolysis, which may become lethal in both children and adults [40]. In experimental studies, we have previously shown an increase in pathological spontaneous EMG activity with histopathological evidence of muscle membrane lesions and apoptotic myonuclei in the limb muscles of mechanically ventilated pigs sedated for five days with both low- and high-dose propofol [41]. It is therefore suggested that the early increase in spontaneous EMG activity among the ICU patients in this study is related to the widespread use of propofol in modern critical care.

There was no significant correlation between preferential myosin loss and the duration of mechanical ventilation, varying between 5 and 165 days, in the 142 patients included in this study. This is in sharp contrast to the progressive preferential myosin loss in response to the duration of immobilization and mechanical ventilation at durations longer than five days reported in an experimental rodent ICU model [15]. It is suggested that the lack of a correlation between myosin loss and the duration of immobilization and mechanical ventilation in this study is related to the retrospective design of the study, emphasizing the need for prospective studies. We are therefore currently conducting a prospective longitudinal study in neuro-ICU patients exposed to 12 days of immobilization and controlled mechanical ventilation. To date, we have followed 12 patients with six repeated biopsies during the 12-day observation period and all patients displayed progressive preferential myosin loss after five days of immobilization and mechanical ventilation, in accordance with previous experimental studies [15]. The lack of a correlation between myosin loss and the duration of mechanical ventilation in the current prospective study may be related to differences in the recovery from myosin loss while still being mechanically ventilated. The mechanisms triggering the recovery from CIM with reduced myofibrillar protein degradation and improved transcriptional regulation of myosin synthesis are very interesting, but they remain elusive and are beyond the scope of the current methodological study.

CIM is characterized by both a decreased muscle membrane excitability (decreased CMAP amplitude) and a preferential loss of myosin and myosin-associated proteins. Both decreased CMAP amplitude and myosin loss were observed among the CIM patients in this study. However, CMAP amplitude and myosin loss were not closely related, indicating differences in the temporal patterns of changes in membrane excitability and myosin expression. In a previous study from our group [14], seven neuro-ICU patients were followed longitudinally for 7–11 days during mechanical ventilation and immobilization; all of them developed CIM with preferential myosin loss at the end of the study [14]. One of the patients was followed longitudinally with ENeG, showing a variable but insignificant change in CMAP amplitude in spite of a decreased myosin:actin ratio (1.3) after 10 days of immobilization and mechanical ventilation, confirming these differences in the temporal patterns, as well as indicating that myosin loss precedes decreased muscle membrane excitability (Figure 9).

In conclusion, conventional electrophysiological methods are sensitive enough to identify the peripheral origin of acquired quadriplegia in ICU patients, but do not reliably distinguish between neurogenic vs. myogenic origins of paralysis. The novel electrophysiological methods presented herein, comparing direct vs. indirect muscle stimulation and refractoriness, represent attractive diagnostic methods, but they are time-consuming, technically demanding, and the precision of CIM diagnosis is questionable. However, decreased indirect vs. direct muscle stimulation strongly suggests a neurogenic lesion. The hallmark of CIM, preferential myosin loss, can be reliably evaluated in small samples obtained with a microbiopsy instrument. The major advantage of this method is that it is less invasive than conventional muscle biopsies, reducing the risk of bleeding in ICU patients, who frequently receive anticoagulant treatment, and it can be repeated multiple times during follow up for monitoring purposes.

## Figures and Tables

**Figure 1 diagnostics-10-00966-f001:**
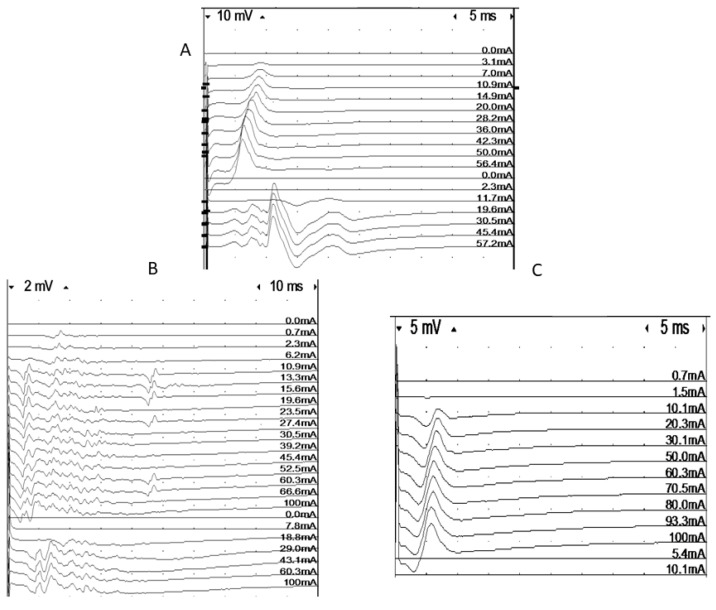
Intramuscular recordings of responses elicited using muscle (dm) and nerve (ne) stimulation in three patients. **A** = normal, **B** = low-amplitude responses for both direct and indirect stimulation, **C** = discrepancy between direct and indirect response amplitude, indicating neuropathy.

**Figure 2 diagnostics-10-00966-f002:**
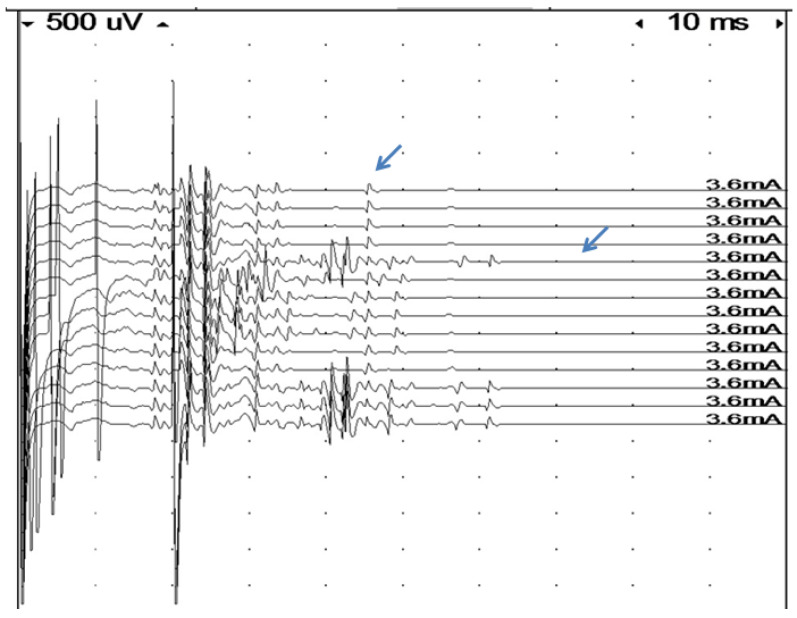
Assessment of refractory period in a very complex signal by double muscle stimulation (high stimulus artifacts indicate time of first stimulation at the beginning of the sweep, and of the second stimulation). The first four traces show single stimulation with a very time-dispersed response and late components. As soon as an extra stimulus is delivered, at an interval of 20 msec, the last component arrows (probably a single fiber action potential), disappear, indicating a very long refractory period in this fiber.

**Figure 3 diagnostics-10-00966-f003:**
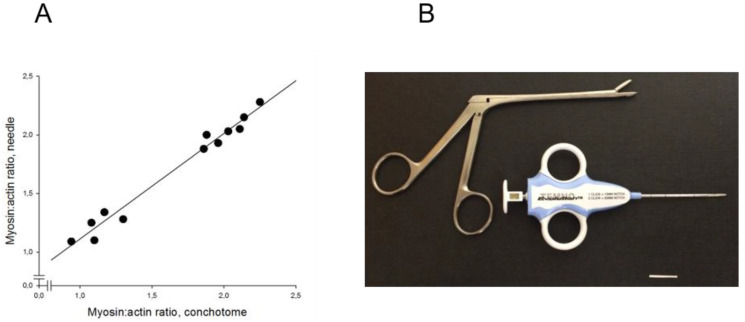
(**A**) M:A ratios determined in 12 intensive care unit (ICU) patients with varying myosin:actin ratios. Ratios compared between samples obtained with the soft-tissue semi-automatic needle and the conchotome instrument (R^2^ = 0.98, *p* < 0.001); (**B**) soft-tissue semi-automatic needle and the conchotome instrument. The horizontal bar denotes 20 mm.

**Figure 4 diagnostics-10-00966-f004:**
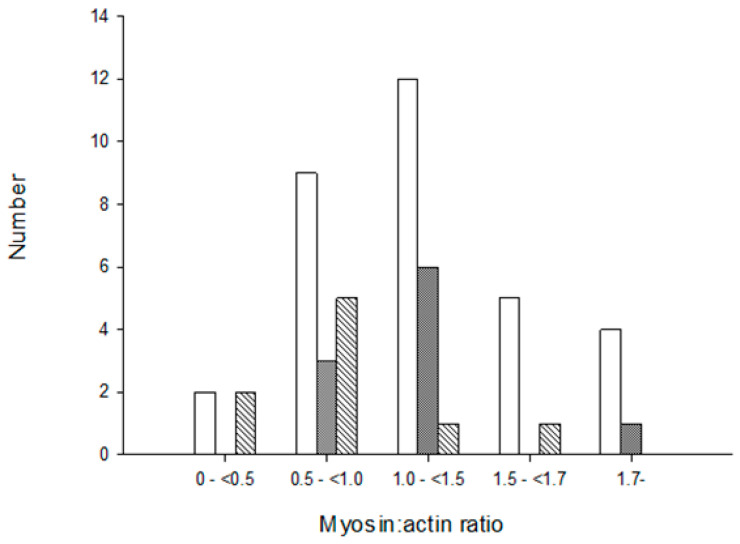
Diagnosis of critical illness myopathy (CIM) based on myosin loss determined from enzyme histochemical and immunocytochemically stained muscle biopsy cross-sections from ICU patients. Open bars represent no CIM, grey bars indicate suspect CIM and diagonal bars represent CIM. The myosin:actin ratios measured in biopsies from the same occasion and muscle are presented in a total of 51 patients in groups separated into severe (<0.5), high (0.5–<1.0), moderate (1.0–<1.5) to small (1.5–<1.7) preferential myosin loss as well as in patients with normal M:A ratios (>1.7).

**Figure 5 diagnostics-10-00966-f005:**
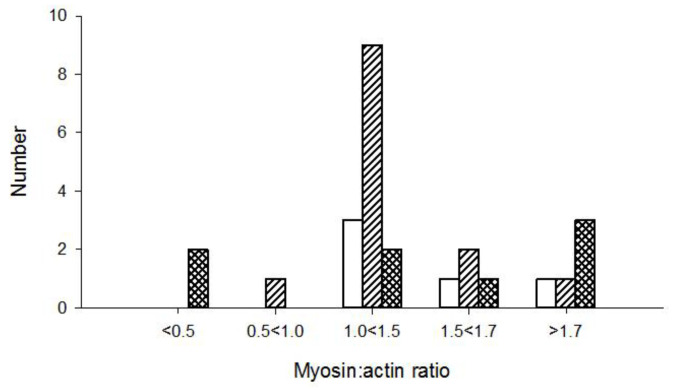
Patients diagnosed as critical illness polyneuropathy (CIP; open bar), normal (diagonal bar), and CIM (hatched bar) based on direct vs. indirect stimulation in patients with different myosin:actin ratios.

**Figure 6 diagnostics-10-00966-f006:**
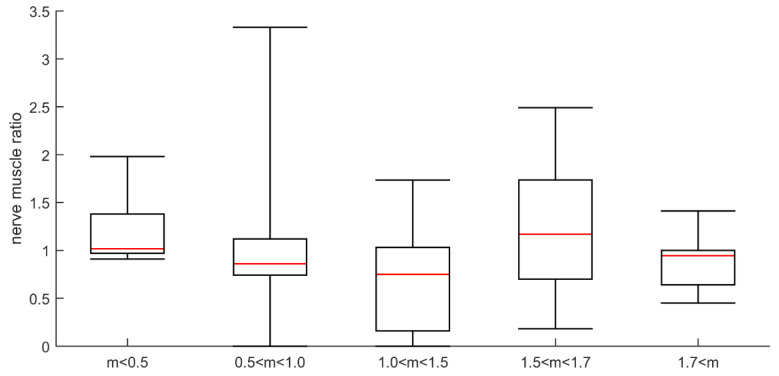
Plot of ratio between amplitude of direct and indirect responses for different myosin:actin groups. The myosin:actin ratios measured in biopsies from the same occasion and muscle are presented in a total of 29 patients in groups separated into severe (<0.5), high (0.5–<1.0), moderate (1.0–<1.5) and small (1.5–<1.7) preferential myosin loss as well as in patients with normal M:A ratios (>1.7).

**Figure 7 diagnostics-10-00966-f007:**
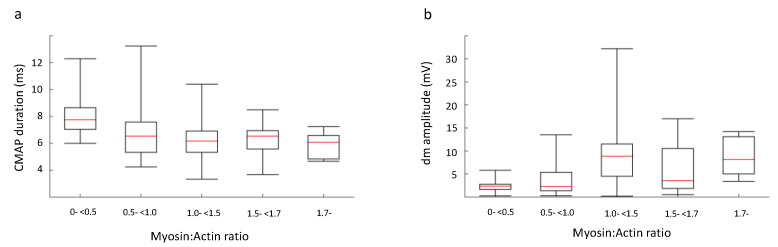
The patients were classified into five groups depending on M:A ratio, as described in Methods. We examined the difference between the five groups in relation to **a**) compound muscle action potential (CMAP) duration, measured using nerve stimulation and surface muscle electrodes, and **b**) CMAP amplitude measured intramuscularly. The differences were compared with a one-way ANOVA and significant differences (*p* < 0.01) were found for CMAP amplitude elicited using direct muscle stimulation. Individual analysis between the five groups revealed a significant difference between m-<0.5 and 1.0-<1.7 of *p* < 0.05 and between m-<0.5 and <1.7 of *p* < 0.01.

**Figure 8 diagnostics-10-00966-f008:**
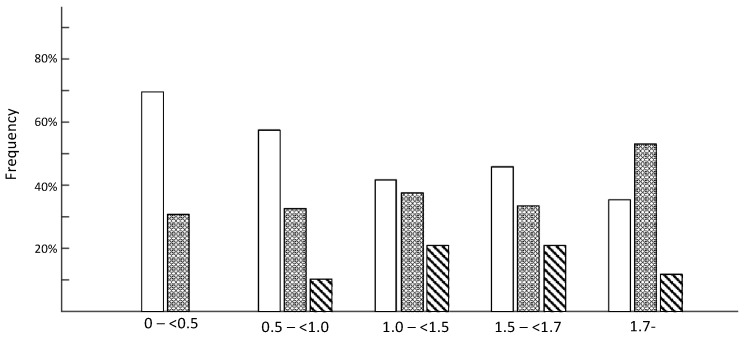
The patients were classified into five groups depending on M:A ratio, as described in Methods. We stratified the patients in each group using the EDX estimates according to the following rules: CMAP index less than −3 (white), CMAP index and SNAP index less than −3 (hatched gray) and others (diagonal black). This EDX stratification can estimate, using EDX data, the presence of myogenic or neurogenic involvement. The presence of myogenic findings according to EDX is seen to increase as the myosin concentration declines.

**Figure 9 diagnostics-10-00966-f009:**
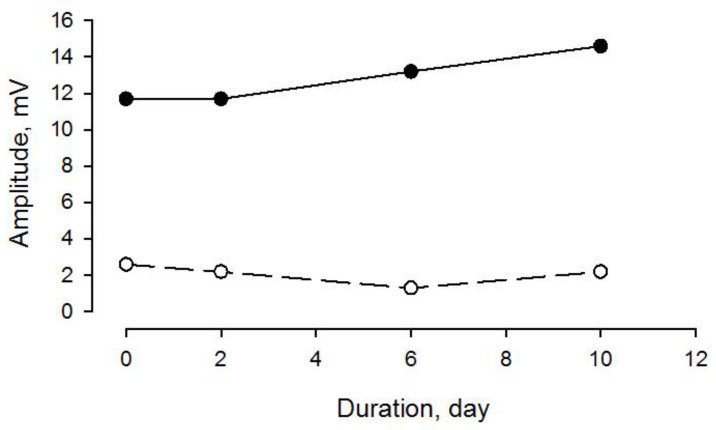
CMAP amplitude upon tibial nerve (filled dots and solid line) and peroneal nerve (open symbol and broken line) supramaximal stimulation and EMG recordings from abductor hallucis (AH) and extensor digitorum brevis (EDB) muscles, respectively, in a neuro-ICU patient exposed to controlled mechanical ventilation for 10 days. After 10 days of ventilation, preferential myosin loss (M:A 1.3) was observed in a distal hind limb muscle (m. tibialis anterior).

**Table 1 diagnostics-10-00966-t001:** The spectrum of main clinical diagnoses are presented for the 142 ICU patients. Two patients were not included in this table since they were excluded because of very high age which may impact on the myosin:actin (M:A) ratio. It should be observed that some patients were suffering from multiple diagnoses. Values are given as ranges and means ± S.D.

TOTAL	142	
Male; number	81	
Female; number	61	
Male; age,	60.1 ± 16.2	9–85
Female; age,	63.8 ± 12.5	26–80
Days since onset of symptoms (days)	29.0 ± 23.0	5–170
Time in respirator (total), mean, SD, range	35.5 ± 28.2	5–165
DIAGNOSIS		
Infection	74	
MOD	26	
Cerebral insult	18	
Cancer	14	
Neurological disorder and others	12	
Trauma	8	
Transplantation	8	

**Table 2 diagnostics-10-00966-t002:** Summary of age, anthropometric, gender and electrophysiological data in the patient material. Number of patients, number of patients with values outside defined confidence limits and % of patients with findings outside defined limits are given. Values are means ± SD.

	Mean	Patients	Outliers	% Outliers	Controls
Age (yrs)	62 ± 14	142	na	na	
Height (cm)	173 ± 10	142	na	na	
Disease duration when examined (d)	27 ± 17	142	na	na	
Spontaneous EMG activity	2.02 ± 1.16	142	131	91	
EMG findings	1.07 ± 1.34	142	na	na	
mCMAP (mV)	6.70 ± 5.90	81	33	41	20.8 ± 7.3
nCMAP (mV)	5.11 ± 4.70	81	61	75	21.1 ± 5.4
mn-ratio	0.94 ± 0.61	71	14	19	1.07 ± 0.2
Refractory period (ms)	10.10 ± 6.77	49	43	88	
Signal duration (ms)	25.05 ± 1.89	33	29	88	8.2 ± 2.4
Myosin:actin ratio (M:A)	1.13 ± 0.47	142	125	87	2.0 ± 0.4
CMAP index	−6.27 ± 3.68	142	122	85	0 ± −1.5
MCV index	−2.93 ± 3.51	100	30	30	0 ± −1.5
SNAP index	−0.89 ± 8.65	142	55	39	0 ± −1.5
SCV index	−0.04 ± 1.66	100	5	5	0 ± −1.5
Median nerve CMAP duration	6.17 ± 1.85	114	31	27	5.03 ± 0.68
Ulnar nerve CMAP duration	6.60 ± 1.69	121	45	37	5.14 ± 0.61
Median nerve CMAP duration (z)	1.92 ± 2.50	114	31	27	0 ± 2
Ulnar CMAP duration (z)	2.78 ± 3.34	121	45	37	0 ± 2

**Table 3 diagnostics-10-00966-t003:** Spontaneous electromyographic (EMG) (fibrillation potentials and positive sharp waves) and motor unit potentials (mups) in patient subgroups divided in relation to preferential myosin loss.

Myosin:Actin Ratio	<0.5	0.5≤ m <1.0	1.0 ≤ m < 1.5	1.5 ≤ m< 1.7	≥1.7
Spontaneous EMG activity (% of pat)	90.1	87.3	87.2	90.9	100.0
Myopathic mups	6 (46%)	12 (30%)	15 (31%)	4 (17%)	3 (16%)
Neuropathic mups	1 (14%)	2 (29%)	3 (43%)	0	1 (14%)
Mixed myo- and neuropathic mups	0	0	2 (33%)	3 (50%)	1 (17%)
Normal mups	0	6 (35%)	7 (41%)	1 (6%)	3 (18%)
No voluntary activation	4 (6%)	18 (29%)	19 (30%)	14 (22%)	8 (13%)

**Table 4 diagnostics-10-00966-t004:** Patients were classified into five groups according the preferential myosin loss, i.e., the M:A ratio. Differences in EDX and clinical parameters between the five groups were investigated using a one-way ANOVA. Value are means (or medians when stated) ± SD.

	0≤ M:A < 0.5	0.5≤ M: < 1.0	1.0≤ M:A < 1.5	1.5 ≤ M:A < 1.7M:A≥	1.7 < myoact	ANOVA
Age (yrs.)	60.9± 10.8	63.3± 17.6	62.0± 11.9	61.2± 16.3	63.4± 10.3	
Height (cm)	170.0± 8.1	171.0± 11.6	174.2± 9.7	171.7± 9.1	174.9± 6.7	
Disease duration when examined (d)	26.0± 21.2	29.4± 18.1	24.5± 12.2	29.3± 20.8	25.9±12.1	
Spont. EMG activity	1.8± 1.0	2.0± 1.3	2.1± 1.3	1.9± 1.0	2.2±0.9	
EMG findings	0.9± 0.9	1.1± 1.4	1.1± 1.4	0.9± 1.3	1.2± 1.6	
mCMAP (mV)	2.5± 1.7	4.0± 3.8	9.3± 6.9	5.8± 5.3	8.7± 4.4	*p* < 0.01
nCMAP (mV)	2.6± 1.7	3.7± 3.6	6.0± 6.0	5.5± 4.3	6.7± 2.1	
mn-ratio	1.2± 0.4	1.0± 0.8	0.7± 0.5	1.2± 0.6	0.9± 0.3	
Refractory period (ms)	14.7± 8.2	11.0± 6.9	9.5± 6.5	9.5± 6.2	3.0± 2.8	
Signal duration (ms)	34.2± 13.4	27.2± 16.1	22.0± 17.2	24.4± 17.9	22.0± 6.0	
CMAP index	−6.4± 2.8	−6.4± 3.0	−6.8± 4.6	−5.3± 3.6	−5.7± 3.1	
MCV index	−2.1± 2.4	−2.9± 3.8	−3.1± 3.8	−2.5± 3.2	−3.7± 3.4	
SNAP index	−1.6± 2.1	−1.3± 2.6	−1.9± 2.1	2.9± 20.3	−2.0± 3.3	
SCV index	0.3± 1.4	0.2± 1.7	−0.3± 1.6	0.1± 1.4	−0.2± 2.1

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
