# Peer review of "Neurogenic vs. Myogenic Origin of Acquired Muscle Paralysis in Intensive Care Unit (ICU) Patients: Evaluation of Different Diagnostic Methods"

_diagnostics, 2020, doi:10.3390/diagnostics10110966_

Round 1

Reviewer 1 Report

The authors aim at evaluating different diagnostic methods for (differential) diagnosis of CIM and CIP in critically ill patients requiring long-term mechanical ventilation. They retrospectively analyzed data and biological samples of 142 ICU patients with acquired quadriplegia/intensive care unit acquired weakness. All Patients were examined with routine electrophysiological methods and biochemical analyses of myosin:actin ratios of muscle biopsies, and in a subset of these patients additional EMG studies and histopathological analyses were performed. The authors found that i) myosin:action ratios from sample by a microbiopsy instrument were identical to samples from standard muscle biopsies, ii) correlations between CMAP properties and myosin:actin ratios were low, and iii) that advanced electrophysiological methods measuring refractoriness and comparing CMAP amplitude after indirect nerve vs. direct muscle stimulation did not increase precision in the diagnosis of CIM. They conclude that i) conventional electrophysiological methods are not appropriate to differentiate between neurogenic and myogenic origin of paralysis/weakness and ii) that myosin loss can be reliably evaluated using a microbiopsy instrument.

This is a very important study featuring a common complication of modern ICU treatment. The current diagnostic criteria for CIM and CIP were proposed by CF Bolton in 2005 and slightly adapted in 2011 by N Latronico and CF Bolton. These criteria demand complex electrophysiological examinations, which sometimes are very challenging on ICU because of e.g. electrical noise and edema. As a consequence the term “ICU-acquired weakness” is increasingly used. For diagnosis of ICU acquired weakness only patient history and clinical examination are required. Since prognosis of CIM and CIP differ, appropriate differential diagnosis is mandatory. Furthermore, exact diagnosis is also important for future clinical trials. Hence the term ICU-acquired weakness should be avoided.

In this view, the observations and findings of the present study are very interesting and clinically relevant. However, the manuscript needs revising in some aspects:

  • Introduction:
    • The introduction is too long and sometimes difficult to follow and contains content repetitions, and should therefore be shortened (e.g. the paragraph about COVID-19 can be omitted).
    • The current diagnostic criteria for CIP, CIM and ICU-acquired weakness should be appropriately cited.
    • The study aims are not sufficiently clear formulated.
  • Methods:
    • The study design (retrospective/prospective) as well as the inclusion/exclusion criteria should be appropriately stated. Additionally it would be interesting to know, i) after how many days of ICU stay/mechanical ventilation the patients were tested and ii) which patients received the additional examinations (e.g. open biopsy, measurement of refractory period…).
    • Figure 1 is important for the study results and should be either moved to the result section (if patients of this study are shown) or to supplementary data, if this analysis war performed in an independent ICU patient cohort.
    • It should be specified which statistical analyses were used for which data and purposes.
  • Results:
    • The results section sometimes contains information that should be mentioned already in the methods section or in supplementary data (e.g. how the muscle biopsies/analyses were conducted, how patients were classified into groups etc.).
    • How many patients fulfilled the diagnosis of CIP, CIM or the combination of both based on the standard diagnostic criteria from Latronico and Bolton (2011)? Since a subgroup of 51 patients underwent histopathological examination, this analysis should be possible. Furthermore, the authors should investigate the validity of the myosin:actin ratio compared to these standard diagnostic criteria. Maybe they even can report sensitivity and specificity of the myosin:actin ratio for diagnosis of CIP and CIM.
    • Table 2 should be moved to supplementary data since the study did not involve healthy subjects according to the study purpose. The same is true for the results of the mentioned pig study and the paragraph about aspiration biopsies.
  • Discussion:
    • The discussion is again very long and contains a lot of new information that would be better placed in the introduction but again also content repetition. The discussion should only summarize the results and then discuss/interpret the findings in relation to previous literature, clinical impact/relevance and future studies.
      •  

      The paragraph stating a dramatic increase of CIM in COVID-19 patients should be deleted or appropriately

Author Response

We are grateful to the reviewers for his/her positive constructive criticisms and suggestions. The manuscript has been rewritten in accordance with suggestions. This has significantly improved our manuscript. Please find comments below.

The authors aim at evaluating different diagnostic methods for (differential) diagnosis of CIM and CIP in critically ill patients requiring long-term mechanical ventilation. They retrospectively analyzed data and biological samples of 142 ICU patients with acquired quadriplegia/intensive care unit acquired weakness. All Patients were examined with routine electrophysiological methods and biochemical analyses of myosin:actin ratios of muscle biopsies, and in a subset of these patients additional EMG studies and histopathological analyses were performed. The authors found that i) myosin:action ratios from sample by a microbiopsy instrument were identical to samples from standard muscle biopsies, ii) correlations between CMAP properties and myosin:actin ratios were low, and iii) that advanced electrophysiological methods measuring refractoriness and comparing CMAP amplitude after indirect nerve vs. direct muscle stimulation did not increase precision in the diagnosis of CIM. They conclude that i) conventional electrophysiological methods are not appropriate to differentiate between neurogenic and myogenic origin of paralysis/weakness and ii) that myosin loss can be reliably evaluated using a microbiopsy instrument.

This is a very important study featuring a common complication of modern ICU treatment. The current diagnostic criteria for CIM and CIP were proposed by CF Bolton in 2005 and slightly adapted in 2011 by N Latronico and CF Bolton. These criteria demand complex electrophysiological examinations, which sometimes are very challenging on ICU because of e.g. electrical noise and edema. As a consequence the term “ICU-acquired weakness” is increasingly used. For diagnosis of ICU acquired weakness only patient history and clinical examination are required. Since prognosis of CIM and CIP differ, appropriate differential diagnosis is mandatory. Furthermore, exact diagnosis is also important for future clinical trials. Hence the term ICU-acquired weakness should be avoided.

In this view, the observations and findings of the present study are very interesting and clinically relevant. However, the manuscript needs revising in some aspects:

  • Introduction:
  • The introduction is too long and sometimes difficult to follow and contains content repetitions, and should therefore be shortened (e.g. the paragraph about COVID-19 can be omitted).

1) During the COVID-19 pandemic spring 2020, we have observed a dramatic (more than10-fold) increase in ICU patients with CIM and preferential myosin loss in our hospital in response to long-term controlled mechanical ventilation. This is an observation shared by colleagues at other university hospitals in Sweden. We therefore believe this justifies the mentioning of COVID-19 patients and the increased number of ICU patients with CIM is valid.

2) Indirect both experimental as well as clinical studies from our group show that the mode of mechanical ventilation, concomitant lung injury, release of factors have negative consequences on peripheral organs including muscle, making the ventilator induced lung injury a possible primary pathophysiological mechanism underlying CIM. Although rarely acknowledged, we think this important information and is currently being explored in our lab directly in experimental studies.  

  • The current diagnostic criteria for CIP, CIM and ICU-acquired weakness should be appropriately cited.

We thank you for this comment and more recent publications reviewing the diagnosis of CIM, CIP and ICU-acquired weakness have been added to the Introduction.

  • The study aims are not sufficiently clear formulated.

We have made an attempt to make this more clear in the revised version of the ms..

  • Methods:
  • The study design (retrospective/prospective) as well as the inclusion/exclusion criteria should be appropriately stated. Additionally it would be interesting to know, i) after how many days of ICU stay/mechanical ventilation the patients were tested and ii) which patients received the additional examinations (e.g. open biopsy, measurement of refractory period…).

It is now stated that this is a retrospective study, the duration at which the patients were tested are given in Table 5. To avoid misunderstandings, this has been better clarified in the table. Open biopsies were not taken from any patient (only in the porcine experiments) and conchotome and/or microbiopsies were taken from all patients. The time of examination for direct vs indirect stimulation and refractory measurements are presented in Table 2 (also clarified that this duration represents the time of examination).

  • Figure 1 is important for the study results and should be either moved to the result section (if patients of this study are shown) or to supplementary data, if this analysis war performed in an independent ICU patient cohort.

The results from Fig. 1 are based on the current patient population and it has been moved to the Results section as suggested.

  • It should be specified which statistical analyses were used for which data and purposes.

This information has been added in the revised version.

  • Results:
  • The results section sometimes contains information that should be mentioned already in the methods section or in supplementary data (e.g. how the muscle biopsies/analyses were conducted, how patients were classified into groups etc.).

Parts of the Results section has been moved to the Methods section as suggested.

  • How many patients fulfilled the diagnosis of CIP, CIM or the combination of both based on the standard diagnostic criteria from Latronico and Bolton (2011)? Since a subgroup of 51 patients underwent histopathological examination, this analysis should be possible. Furthermore, the authors should investigate the validity of the myosin:actin ratio compared to these standard diagnostic criteria. Maybe they even can report sensitivity and specificity of the myosin:actin ratio for diagnosis of CIP and CIM.

The Diagnostic algorithm suggested by Latronico and Bolton 2011

Will separate between neuromuscular transmission disorders and CIP/CIM, but will not give a reliable separation between CIM and CIP based on CMAP duration, direct muscle stimulation and “consider” muscle biopsy. I assume the authors are considering histopathological evaluation. If this is the case, then it should be obvious by looking at Figures 4, 5, 6, 7 and 8 that this will not give a reliable diagnostic precision in the separation between CIM and CIP. In a current prospective study currently being undertaken in our group where 12 neuro-ICU patients have been exposed to controlled mechanical ventilation for 12 days and where conchotome and microbiopsies were taken 6 times during the observation period, all 12 patients showed a progressive preferential myosin loss, severe muscle fiber loss and decline in force generation capacity, i.e., the characteristics of CIM. However, in none of these patients were there a significant change in CMAP amplitude measured at the same time as biopsies were taken. Thus, it is difficult to see how these standard diagnostic criteria will impact on the sensitivity/specificity of the myosin:actin ratio. In my experience conducting myosin:actin ratios in clinical samples during the past 25 years, I have only observed a preferential myosin loss in one patient without CIM. This patients had developed paraplegia in response to lung cancer and cancer cachexia which is in accordance with experimental cancer cachexia models. However, the differential diagnosis between the two conditions are not difficult. When the results from the cancer cachexia patients were submitted to a clinical journal, the reviewer claimed that this was a typical case of CIM in spite of the fact the patient had not been critically ill, ICU treated or mechanically ventilated!

  • Table 2 should be moved to supplementary data since the study did not involve healthy subjects according to the study purpose. The same is true for the results of the mentioned pig study and the paragraph about aspiration biopsies.

As suggested, Table 2 has been moved to supplementary data.

  • Discussion:
  • The discussion is again very long and contains a lot of new information that would be better placed in the introduction but again also content repetition. The discussion should only summarize the results and then discuss/interpret the findings in relation to previous literature, clinical impact/relevance and future studies.

The Discussion has been shortened as suggested and is now less than two text pages..

The paragraph stating a dramatic increase of CIM in COVID-19 patients should be deleted or appropriately

We have observed a dramatic increase in ICU patients diagnosed with CIM during the pandemic spring 2020 and this information has been added to the Introduction and deleted from the Discussion.

Reviewer 2 Report

The manuscript of Humberto Gonzales Marrero and coworkers describes the electrophysiological findings in patients with critical illness together with skeletal muscle weakness/paralysis and aims to distinguish between critical illness myopathy (CIM) and critical illness polyneuropathy (CIP). The authors show that conventional electrophysiological examination cannot reliably distinguish between CIM and CIP, which they set out to test. The idea is scientifically sound and of clinical importance as tool to determine the underlying cause of the muscle weakness in the ICU-patient. Of interest was that the amplitude of compound muscle action potential using direct muscle stimulation was decreased in the ICU-group with the lowest M:A ratio suggesting severe CIM. Additionally, they show that small muscle biopsies acquired with soft-tissue semi-automatic needle is similar to biopsies obtained by the standard conchotome when comparing Myosin-to-Actin ratio (M:A). This finding lessens the invasiveness of muscle biopsy acquisition.

The data appears solid and the study is carried out by experts in the field for both the electro-physiological examinations and the molecular experiments.

The language is sometimes a bit convoluted but overall grammatical correct. However, the authors need to carefully check the spelling and spacings in the paper. The spelling errors are numerous but appears to be primarily in the Materials & Methods, Results and Figures and Tables legends. I listed hopefully most of them at the end.

Was there a correlation between M:A ratio and mortality and/or weaning failure?

One possibility is that the results of the study could been clearer if they used regression analysis with scatter plots using Myosin-to-Actin ratio (M:A) as x-axis and the electrophysiological parameters as y-axis instead of grouping the M:A. It could be interesting to do these comparisons for mCMAP, nCMAP, refractory period, signal duration, CMAP index and SNAP index from table 5. How does a regression analysis come out if the groups from figure 8 is used?

The authors suggest that underlying neuropathies is present but did not specify, so had any of the ICU patients previously been diagnosed with any neuropathies? If so, how well did it match the histological/immunohistological findings?

What was the stratification for figure 8 (CMAP and SNAP index <-3, SNAP index <-3 or indices within reference values) based on? Can the authors briefly expand on this?

Minor comments:

Statistics: Was any post hoc test used for the one-way ANOVA? Line 282: The statistics were calculated not estimated, I assume.

For Sensory Nerve Action Potential Index calculation, what does ‘dig III’ refer to? Lines 220-223

ENeG is not included on the abbreviation list

Table 1: Time in respirator (total), mean, SD, range – no unit is indicated, I assume days

Table 2: What was the mean age of the healthy controls? What does 10, 20 and 25% refer to? Percentile?

For table 3: Can the reference values be included as a column it would it make it easier to judge how off the outliers were.

Figure 2: It is difficult to read the stimulation current primarily for A and B panel. Can readability be improved?

For Figure legend 4, can it be highlighted that these data comes from ICU-patients.

Figure 5: Can the N-values be added for CIM, CIP and normal groups be added?

Spelling errors

Line 26: responmse

Line 109: straing

Line 139: givn, mens

Line 178: Qiaggen

Line 250: reported..

Line 255: thress

Line 256: discrepacne and nauropathy

Line 258: patients.To

Line 267: cprresponding

Line 292: seprates

Line 344:  binstruments

Line 356: ptients

Line 357: CIM.Tthe

Line 405: stimulatiomn

Line 408: imndirect

Line 424: Spontaneoue and potantials

Line 438: dirct

530: myosfibrillar

550: a neuro-ICU patients -> patient

Author Response

We are grateful to reviewer for positive and constructive criticism. Also, we apologize for the numerous misspellings. We have corrected the ones identified by reviewer and made additional corrections in the text. All corrections are identified by track changes. Dig III refers to digitorum III or the long finger and dig V same as little finger.

Comments and Suggestions for Authors

The manuscript of Humberto Gonzales Marrero and coworkers describes the electrophysiological findings in patients with critical illness together with skeletal muscle weakness/paralysis and aims to distinguish between critical illness myopathy (CIM) and critical illness polyneuropathy (CIP). The authors show that conventional electrophysiological examination cannot reliably distinguish between CIM and CIP, which they set out to test. The idea is scientifically sound and of clinical importance as tool to determine the underlying cause of the muscle weakness in the ICU-patient. Of interest was that the amplitude of compound muscle action potential using direct muscle stimulation was decreased in the ICU-group with the lowest M:A ratio suggesting severe CIM. Additionally, they show that small muscle biopsies acquired with soft-tissue semi-automatic needle is similar to biopsies obtained by the standard conchotome when comparing Myosin-to-Actin ratio (M:A). This finding lessens the invasiveness of muscle biopsy acquisition.

The data appears solid and the study is carried out by experts in the field for both the electro-physiological examinations and the molecular experiments.

Dear Reviewer,

We are grateful for the very positive and constructive criticism. We have corrected the ms. in accordance with your advice. This has improved our ms considerably.

The language is sometimes a bit convoluted but overall grammatical correct. However, the authors need to carefully check the spelling and spacings in the paper. The spelling errors are numerous but appears to be primarily in the Materials & Methods, Results and Figures and Tables legends. I listed hopefully most of them at the end.

we apologize for the numerous misspellings. We have corrected the ones identified by reviewer and made additional corrections in the text. All corrections are identified by track changes.

Was there a correlation between M:A ratio and mortality and/or weaning failure?

In those patients were we can track survival there is no difference in those alive (myosin:actin ratio 1.12 ± 0.47, n=69) and those who are dead (1.07 ± 0.47, n=48), p=0.6. This is in accordance with an ongoing study with Leuven University where we could not see a significant relationship between survival and the myosin:actin ratio only a trend towards an improved survival and a lower myosin:actin ration. Due to the wide spread in age, underlying disease and preclinical history, it is argued if this type of comparison is meaningful in the heterogenous populations studied.

One possibility is that the results of the study could been clearer if they used regression analysis with scatter plots using Myosin-to-Actin ratio (M:A) as x-axis and the electrophysiological parameters as y-axis instead of grouping the M:A. It could be interesting to do these comparisons for mCMAP, nCMAP, refractory period, signal duration, CMAP index and SNAP index from table 5. How does a regression analysis come out if the groups from figure 8 is used?

We have added these scatterplots as a supplemental figure 2 in the revised ms.

The authors suggest that underlying neuropathies is present but did not specify, so had any of the ICU patients previously been diagnosed with any neuropathies? If so, how well did it match the histological/immunohistological findings?

As mentioned in the ms, enzymehistochemical and immunocytochemical analyses of muscle biopsies have shown fiber type grouping and grouped atrophy in some of the patients included in this study and misinterpreted as a sign of CIP. However, this is a misconception since both grouped atrophy and fiber type grouping are signs of peripheral denervation and reinnervation. A process which precedes the acute acquired paralysis in the ICU patients by multiple months/years and reflects a chronic denervation. The general ICU population typically has a long history of various pathologies including neuropathies such as Diabetes. A study was initiated within our consortium many years ago to study this specifically by screening patients to the ICU with EneG/EMG to quantify the number of ICU patients with a preexisting neuropathy. Unfortunately, this is a study which has stalled, and we do not have this information available at the moment. However, this is a study which will be resumed shortly with a more motivated clinician.

What was the stratification for figure 8 (CMAP and SNAP index <-3, SNAP index <-3 or indices within reference values) based on? Can the authors briefly expand on this?

We now write in the Methods:

Healthy subjects were used to get normative values for the indices and only 5% of the subjects had index values below -3. The stratification of abnormality at -3 resulted in a p-value of 0.05.

Minor comments:

Statistics: Was any post hoc test used for the one-way ANOVA?

We now write:

Tukeys range test was used to identify individual differences between the groups.

Line 282: The statistics were calculated not estimated, I assume.

Corrected

For Sensory Nerve Action Potential Index calculation, what does ‘dig III’ refer to?

Specified.

Lines 220-223

ENeG is not included on the abbreviation list

Specified together with EMG.

Table 1: Time in respirator (total), mean, SD, range – no unit is indicated, I assume days

Added

Table 2: What was the mean age of the healthy controls? What does 10, 20 and 25% refer to? Percentile?

We now write in the legend:

10, 20 and 25% indicate the number of healthy subjects with M:A ratio below the corresponding level.

For table 3: Can the reference values be included as a column it would it make it easier to judge how off the outliers were.

Reference values are now included in the table.

Figure 2: It is difficult to read the stimulation current primarily for A and B panel. Can readability be improved?

An improved figure is included in the revised version of the ms.

For Figure legend 4, can it be highlighted that these data comes from ICU-patients.

This information has been added.

Figure 5: Can the N-values be added for CIM, CIP and normal groups be added?

N-values for ICU patients diagnosed as normal, CIM and CIP are given on the Y-axis.

Spelling errors

All the errors below together with additional misspellings have been corrected.

Line 26: responmse

Line 109: straing

Line 139: givn, mens

Line 178: Qiaggen

Line 250: reported..

Line 255: thress

Line 256: discrepacne and nauropathy

Line 258: patients.To

Line 267: cprresponding

Line 292: seprates

Line 344:  binstruments

Line 356: ptients

Line 357: CIM.Tthe

Line 405: stimulatiomn

Line 408: imndirect

Line 424: Spontaneoue and potantials

Line 438: dirct

530: myosfibrillar

550: a neuro-ICU patients -> patient

Round 2

Reviewer 1 Report

The authors provide a much improved version of their paper and answered my concerns.

To my opinion, the paper has significantly improved.